# Effects of Convective Drying of Quince Fruit (*Cydonia oblonga*) on Color, Antioxidant Activity and Phenolic Compounds under Various Fruit Juice Dipping Pre-Treatments

Hasmet Emre Akman [1], Ismail Boyar [1], Sadiye Gozlekci [2], Onur Saracoglu [3] and Can Ertekin [1,*]

[1] Department of Agricultural Machinery and Technologies Engineering, Faculty of Agriculture, Akdeniz University, Antalya 07070, Turkey
[2] Department of Holticulture, Faculty of Agriculture, Akdeniz University, Antalya 07070, Turkey
[3] Department of Horticulture, Faculty of Agriculture, Tokat Gaziosmanpasa University, Tokat 60240, Turkey
* Correspondence: ertekin@akdeniz.edu.tr

**Abstract:** In this study, 3 mm thick quince slices were dipped in bitter orange (BO), tangerine (*Citrus deliciosa* Mediterranean) (CDM), orange (O), tangerine (*Citrus reticulata* Owari) (CRO), pomegranate (P) juices and a non-pre-treatment (control) dried at an air temperature of 70 °C and air velocity of 1.5 m/s. According to the results of the pre-treatment trials, drying time was found to be a minimum of 121 min in pomegranate at 5 min dipping time to reach a safe moisture content of 10% (w.b.). The lowest total color change (ΔE) values were observed in orange juice dipped samples (10.19). We found the highest total phenolics (TP) (16.77 mg GAE/g d.w.) in CDM, Trolox equivalent antioxidant capacity (TEAC) (32.49 μmol TE/g d.w.) in P and total flavonoid (TF) (2018.15 mg GAE/kg d.w.) in BO for 5 min dipping time values. As a result, pomegranate juice pre-treatment in all dipping times improved the biocompounds for quince slices.

**Keywords:** thermal processing; fruit; pre-treatment; biocompounds

## 1. Introduction

Today worldwide quince cultivation has increased by 13.6% and advanced to over 90,000 hectares in area; China leads with 55.2%, where Iran and Turkey follow with 7.7% and 7.6%, respectively. Efficiency wise, Turkey leads with the maximum product output, regardless of China having a seven times greater cultivated area, followed only by Turkey, then Uzbekistan, Iran and Azerbaijan [1].

Quince fruit is generally used as a marmalade, jelly, jam, puree, fruit juice and food additive due to its acrid, acidic and hard structure [2–4]. Quince, which has a low fat content, is rich in vitamins, mineral substances, organic acids, proteins, carbohydrates, amino acids and fibers [5,6]. In addition, quince is rich in antioxidants and phenolic substances [7,8]. Due to these properties, quince has a protective effect against diseases such as eczema [9], cancer [10] and ulcers [11] and strengthens the heart and brain [8,9,12].

There have been various methods adopted to reduce the moisture content of organic and inorganic substances, thus some of these methods are even applicable industrially. In order to preserve an organic material or food from decomposition, in this case the quince fruit, it is dewatered or dried to maintain a minimum moisture content which would bring the microbiological activity to a halt [13].

Drying is one of the major steps in food processing in order to maintain longevity in food preservation [14] and helps to achieve reduced costs in packaging, storage and transportation [15].

Conventional drying is one of the proven tools in food industry processes. Through the drying process, various types of fruits and vegetables can be efficiently preserved to last longer. Implementing convective drying is a complex thermal course where heat and

mass transfers occur at the same time in temporary conditions. Thus, the temperature of drying air, velocity and relative humidity plays major roles in this process [16].

Sun drying, which is a well-known traditional drying method, can produce poor results in terms of food quality and safety. Countless unwanted results in using such traditional methods in food preservation have led us to improve new technologies such as microwave, vacuum processing, infrared, freezing and implementing hybrid drying methods where more than one of these methods is brought together to reach the desired results [17–19].

Every drying method is subjected to processed food properties as the expected end product conditions, maturity level, available tools and machinery to be implemented and total costs of the entire procedure required to reach the desired results.

Color is a critical parameter to be examined for considering running the right drying method for the specific type of material in specific conditions. Studies with banana, apple, potato and carrot are good examples which finally indicate freezing and osmotic dehydration give much better results in preservation of color rather than applying convective air, vacuum and microwave methods which prove to adversely affect L*, a* and b* color parameters of the mentioned samples [20].

The investigations on various methodologies and chemical applications, as well as their influence on drying processes, are included below. Many studies have utilized various pre-treatments before the drying process, such as dipping into lemon juice, ascorbic acid, salt solution, honey immersion, osmotic pre-treatment, bleaching, cooling, ultrasound, radiation and freezing (steam or water) [21–27].

In this experimental study, the aim was to obtain high quality dried quince fruit slices under different dipping pre-treatments such as bitter orange (*Citrus aurantium*) (BO), tangerine (*Citrus deliciosa* Mediterranean) (CDM), orange (*Navel citrus* Sinensis) (O), tangerine (*Citrus reticulata* Owari) (CRO), pomegranate (*Punica granatum*) (P) juices and non-pre-treatment (control) at convective drying. The aim was also to observe drying time, color properties, total phenols (TP), Trolox equivalent antioxidant capacity (TEAC) and total flavonoids (TF) in convective drying of quince slices after different dipping pre-treatments.

## 2. Materials and Methods

### 2.1. Sample Preparation

Fresh quinces (*Cydonia oblonga*) were obtained from a local market in Autumn 2020 in Antalya, Turkey (Table 1). Samples were preserved at 4 °C in the refrigerator during the experiments (in 12 days). Samples were maintained to reach room temperature prior to the start of the experiments. Quinces were washed with tap water, sliced into 3, 4, 5 and 6 mm thick discs and removed of their seeds by a stainless steel hand device.

**Table 1.** Quince Physical and Color Properties Used in Drying Experiments.

| Fruit Weight (g) | Fruit Volume (cc, mL, cm$^3$) | Fruit Length (mm) | Fruit Diameter (mm) | Fruit Shape Index | L* | a* | b* | Chroma (C*) | Hue (h°) |
|---|---|---|---|---|---|---|---|---|---|
| 461.09 | 565.42 | 111.74 | 95.69 | 0.86 | 72.44 | 9.04 | 65.45 | 66.13 | 82.20 |

### 2.2. Pre-Treatments of Samples

In this study, 6 different fresh fruit juices as pre-treatment were applied to sliced quince samples. These materials were bitter orange juice, 2 different tangerine juices, orange juice, pomegranate juice and a control (Table 2). As a material of juices, the fruits were obtained from the market, squeezed and about 1.5 L was used as it was in the same day. Samples were subjected to 3 repetitions and dipping times of 1, 3 and 5 min and the control (0 min). After the above mentioned pre-treatments, control and treated samples were dried in cabinet convective dryer at 60, 70 and 80 °C drying air temperatures and constant drying air velocity of 1.5 m/s. It could be checked in Table 2, pre-treatments, dipping times and soluble solid contents of fruit juices. The initial moisture content of the quince slices was

determined by oven method at 70 °C temperature until a constant weight was reached [13].

**Table 2.** Quince Pre-Treatments and Dipping Times.

| Acronym | Pre-Treatment | Dipping Time (min) | Soluble Solid Content (Brix) |
|---|---|---|---|
| NP | Non pre-treatment | - | 13.47 (Fresh) |
| BO | Bitter orange (*Citrus aurantium*) juice | 1-3-5 | 11.88 |
| CDM | Tangerine (*Citrus deliciosa* Mediterranean) juice | 1-3-5 | 13.28 |
| O | Orange (*Navel citrus* Sinensis) juice | 1-3-5 | 12.76 |
| CRO | Tangerine (*Citrus reticulata* Owari) juice | 1-3-5 | 11.82 |
| P | Pomegranate (*Punica granatum*) juice | 1-3-5 | 17.95 |

*2.3. Cabinet Convective Dryer*

Drying tests were carried out via cabinet convective dryer, 1000 Watt Dalle LT-27 with horizontal air flow dehydration fan, digitally adjustable drying air temperature between 30–90 °C, 220–240 V/50 Hz, 45–50 dB. The device is comprised of 12 trays with a 14 kg weight stainless steel 462 × 402 × 447 mm body, produced in Turkey.

*2.4. Drying Tests*

The cabinet convective dryer was set to maintain stable operating temperatures for 30 min prior to each test. The tests were carried out at constant drying air temperature of 60, 70 and 80 °C and drying air velocity of 1.5 m/s was applied horizontally to the trays. Samples were weighed by taking the samples out manually for about one minute every 45 min in comparison with initial conditions of the test group. Testing stopped when samples reached the final moisture content of 10% (w.b.). The experiments were carried out in triplicate for each application and mean values were given in the results.

*2.5. Color Measurement*

Device model PCE-CSM3, as CIE L*, a*, b*, C*, h, CIE L*, a*, b* selectable, with 8°/d measurement geometry, Ø8 mm measurement diaphragm and silicone photoelectric diode sensored high precision measurement device produced in Turkey was utilized during colorimetry tests. The light source was LED D65 working with rechargeable Li battery with a dimension of 200 × 70 × 100 mm.

The L* represents the color's brightness, ranging from 0 to 100, while positive a* and positive b* represent red and yellow, respectively. The color saturation and h° values of 0°, 90°, 180° and 270° in the Hunter Lab scale correspond to the colors red, yellow, green and blue, respectively [28]. Equation (1) was used to calculate the total color value difference (ΔE) between the samples. Color measurements acquired from fresh quince slices were used as a reference in this equation ($L_{ref}$, $a_{ref}$, $b_{ref}$). The browning index (BI) was calculated using Equations (2) and (3) [29];

$$\Delta E = \sqrt{\left[ (L^* - L_{ref})^2 + (a^* - a_{ref})^2 + (b^* - b_{ref})^2 \right]} \tag{1}$$

$$BI = [100(x - 0.31)] \div 0.172 \text{ where } x, \tag{2}$$

$$x = (a^* + 1.75L^*) \div (5.645L^* + a^* - 3.012b^*) \tag{3}$$

*2.6. Bioactive Compounds (Phenolic (TP), Trolox Equivalent Antioxidant Capacity (TEAC), Total Flavonoid (TF) Analysis)*

The dried fruits were soaked and homogenized using a food mixer to extract bioactive components. For analysis, the materials were divided into three tubes and kept at −20 °C until analysis (one week). In a food mixer at room temperature, the materials were dissolved and homogenized (21 °C). The resulting slurry was centrifuged at 4 °C for 30 min to separate the juice from the pulp (12,000× $g$). The freshly extracted juice was diluted with distilled

water, divided into numerous sample aliquots, and stored at −20 °C until employed in phenolics, flavonoid and antioxidant analyses (one week).

The total phenolics (TP) content was determined using Singleton and Rossi's method [30]. Shortly, fruit slurries (0.5 g) were extracted by keeping (without shaking) in 19.5 mL buffer containing acetone, water and acetic acid (70:29.5:0.5 *v/v*) for 2 h at dark. Three times the samples were reproduced. The extracted material was mixed with Folin-phenol Ciocalteu's reagent and water for 8 min before being added to a 7% sodium carbonate solution. After 2 h, the absorbance at 750 nm was measured using an automated UV-Vis spectrophotometer (PG Instruments Model T60U). The standard utilized was gallic acid. The data were represented using gallic acid equivalents (GAE) $g^{-1}$ dry weight in miligrams (mg d.w.).

Using the Chang et al. [31] method, total flavonoids were calculated as g QE kg dry weight$^{-1}$ (quercetin equivalent). To calculate total antioxidant capacity, the Trolox equivalent antioxidant capacity (TEAC) method was utilized. The usual TEAC process was used to produce 10 mmol/L ABTS (2.2-azino-bis-3-ethylbenzothiazoline-6-sulfonic acid) in acetate buffer [32]. For better stability, the mixture was diluted to an absorbance of 0.7000 ± 0.01 at 734 nm in an acidic solution of 20 mM sodium acetate buffer (pH 4.5). For spectrophotometric analysis, 2.90 mL ABTS solution was combined with 100 mL fruit extract and incubated for 10 min. The absorbance at 734 nm was calculated after that. The findings are given in mol Trolox equivalent (TE) g dry weight$^{-1}$.

### 2.7. Statistical Analysis

Tests were carried out in 3 repetitions and statistically evaluated through the effects of various pre-treatment methods on drying time and some quality indicators. Statistical analyzes were examined in 3 different groups as fruit juices, chemicals and dosage chemicals. SPSS (Version17; Chicago, IL, USA) statistics software was utilized on the results obtained. Duncan test at $p \leq 0.05$ level was applied for comparison of average values.

## 3. Results and Discussion

### 3.1. Effects of Drying Air Temperature on Drying Time

Quince slices were dried to the equilibrium moisture content in all drying experiments. However, the safe moisture content of the quince slices was accepted as 10% (w.b.). As shown in Figure 1, the temperature increase caused relatively shorter drying times. The difference shows in the figure below how particularly non-pre-treated slices in 3 mm thickness went through as heated from 60 °C to 80 °C. At higher drying air temperatures, the moisture movement in the slice was faster by diffusion. Reaching to the safe moisture content took 170 min at 60 °C. This value decreased to 165 min at 70 °C and to 158 min at 80 °C for a slice thickness of 3 mm. Therefore, the effect of drying air temperature on drying time was found to be statistically significant.

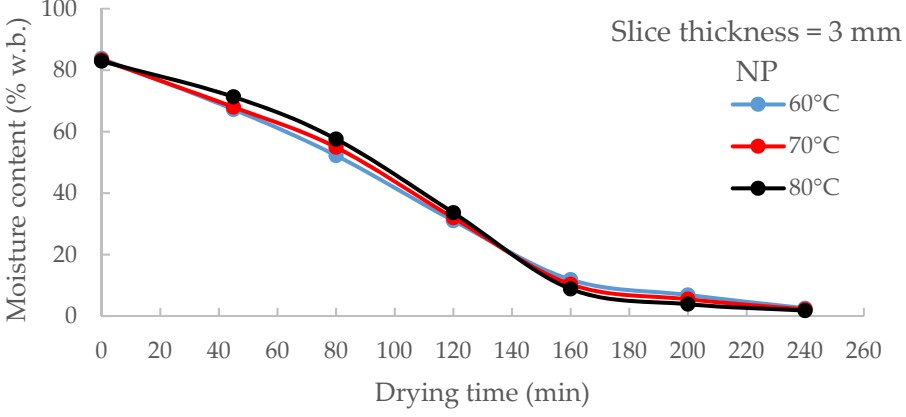

**Figure 1.** Effect of drying air temperature on drying time of non-pre-treated (NP) quince fruit.

Denge [33] discovered that raising the drying air temperature in the oven and vacuum drying the quince reduced the drying time. Microwave drying time was 10 min at 750 W and 240 min at 100 W, hence raising the power level lowered drying time. Aktas et al. [34] found the lowest drying time as 240 min by infrared drying at drying conditions of 40 °C and 1.22 m/s, while the longest drying time was approximately 390 min at 35 °C and 2.45 m/s. Abbaspour-Gilandeh et al. [35] employed convective drying and discovered that at lower drying air temperatures, the overall drying time was longer. Szychowski et al. [36] used convective (CD), vacuum–microwave (VMD), freeze drying (FD) and combination drying (CDVMFD) procedures. The drying time was lowered from 300 to 240 min by raising the air temperature in CD from 50 to 70 °C. The drying time in VMD was reduced from 120 to 34 min by increasing the microwave power from 120 to 480 W. Increasing temperature from 50 to 70 °C reduced the entire drying time in the CDVMFD process from 135 min to 112 min. Celen and Kus [37] found that microwave power level affected the drying process according to the findings; increasing the power level increased the moisture loss and reduced drying time. The drying times were 39, 95 and 57 min at 2000 W and 39, 50 and 36 min at 2800 W, respectively, for the aforementioned belt velocities. Tzempelikos et al. [38] investigated convection drying of 12 mm quince slices. Raising the temperature from 40 °C to 50 °C reduced drying time by about 25%, from 50 °C to 60 °C reduced the drying time by about 36%, and from 40 °C to 60 °C reduced overall drying time by about 54%. Tzempelikos et al. [39] studied convective drying of quince slices at varied drying air temperatures, bulk velocity, drying air velocity and relative humidity. According to the statistics, the drying air temperature had a significant impact on the drying time. Increasing the drying air temperature from 40 to 60 °C cut the overall drying time in half. Karaaslan et al. [40] dried quince slices (5 mm) by microwave–convective dryer. The researchers discovered that increasing the microwave power level lowered the drying time. Drying times were 104, 39, 24, 22 and 12 min at 180, 360, 540, 720 and 900 W microwave power levels, respectively. To dry quince cubes, Izli and Polat [41] utilized convection drying. It took 120, 90, 70 and 50 min at 45, 55, 65 and 75 °C, respectively. As a result, the drying time was significantly reduced as the temperature rose. Kaya et al. [42] dried quince slices (4 mm) at drying air temperatures of 35, 45 and 55 °C, velocities of 0.2, 0.4 and 0.6 m/s, and relative humidities of 40, 55 and 70%. According to the findings, raising the drying air temperature or velocity and decreasing relative humidity lowered overall drying time. When the drying air temperature increased from 35 to 55 °C, the overall drying time was lowered by 55 percent. After osmotic drying by concentrated fruit juice solutions at 40 Brix, Turkiewicz et al. [43] dried quince cylinders by convective drying and found the drying time to be between 360 to 480 min. When the lowest drying temperature was used, the longest drying time was obtained. Turkiewicz et al. [44] evaluated a convective, vacuum–microwave, a combination of convective pre-drying and vacuum-microwave finish drying and freeze drying methods. In convection drying, the drying time varied between 360 and 480 min. At the lowest drying air temperature, the longest drying time was recorded. Furthermore, increasing the temperature in convective drying by 20 °C reduced the drying time by around 25%. Vacuum–microwave drying took anywhere from 28 min at 480 W to 96 min at 120 W. When the microwave power increased from 120 to 480 W, the drying time was lowered by 70%. When the microwave power was increased from 120 to 240 W, the drying time was decreased in half. The results are compatible with the literature.

### 3.2. Effect of Thickness on Drying Time

The fastest drying time was reached in quince slices of 3, 4, 5 and 6 mm, respectively, as a result of drying studies for different slice thicknesses without any pre-treatment at a constant drying air temperature of 70 °C and velocity of 1.5 m/s. The drying times of 3 and 4 mm thick quince slices were quite near to each other, as shown in Figure 2. For slice thicknesses of 3, 4, 5 and 6 mm, the drying time to obtain the desired moisture content was 165, 173, 223 and 260 min, respectively. As a result, the drying time rose as the

slice thickness grew. The movement of the moisture to the surface of the product took a longer time for higher thicknesses. The differences between drying times for different slice thicknesses were also found to be statistically important.

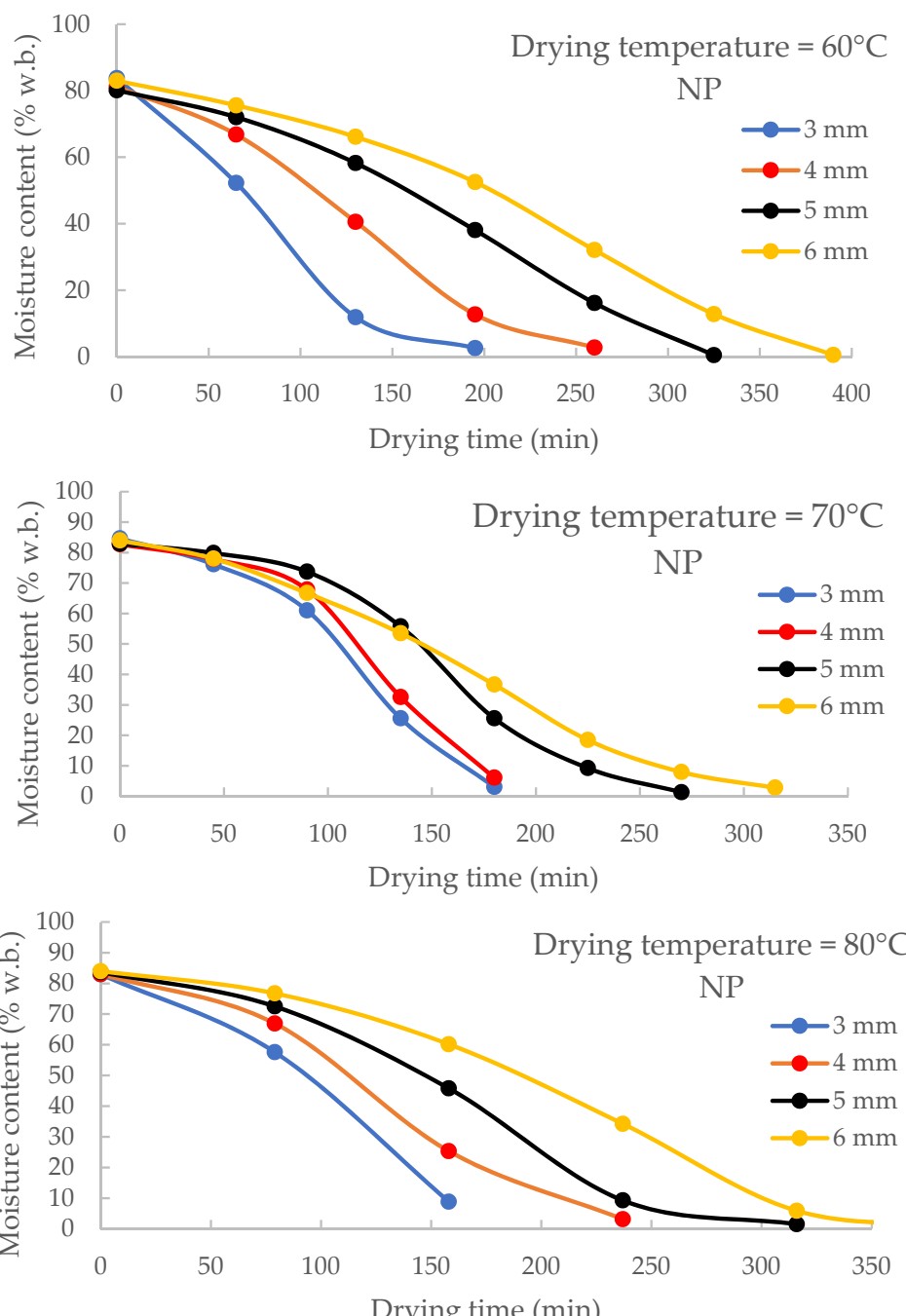

**Figure 2.** Effect of slice thickness on drying time of non-pre-treated (NP) quince fruit at different drying air temperatures.

Denge [31] dried quince slices in the oven at the slice thicknesses of 1, 5 and 10 mm. According to the results, decreasing the slice thickness improved the drying rate in all pre-treatments. The shortest drying time was obtained at a slice thickness of 1 mm. While drying time was 3–4 h for 1 mm, it was 17–19 h for 5 mm and 20–22 h for 10 mm slice thickness. Similar results were also found for different fruit and vegetable slices, including eggplant [45], mango [46] and tomato [47], drying time was shorter for thinner slices.

### 3.3. Effects of Dipping Pre-Treatments on Drying Time

In order to determine the most effective dipping materials for increasing the drying rate, different fruit juices were applied. They generally improved the moisture transfer from the surface.

Figure 3 showed that the longest drying time to reach the desired moisture content was obtained in non-pre-treated (NP) and BO dipped samples for 1 min dipping time. Thus, pre-treatment applications definitely provided shorter drying times in the drying of quince slices except BO. The drying time was found as a minimum pre-treated with P and CDM, in 135 and 136 min, respectively.

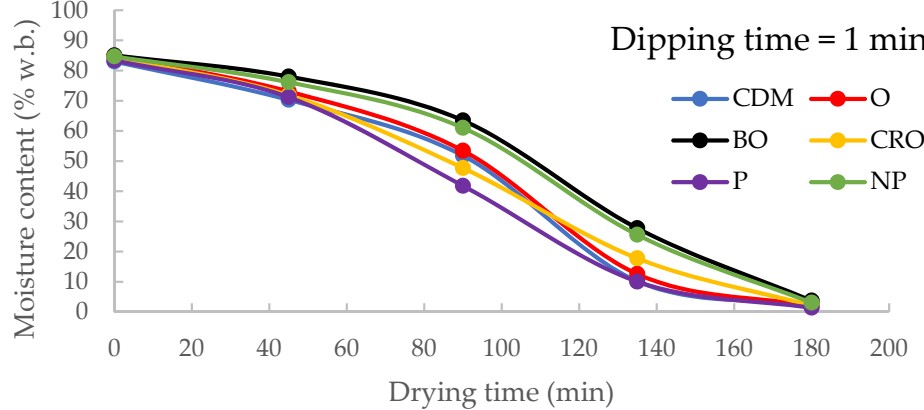

**Figure 3.** Effect of 1 min dipping time on drying time of different pre-treatments (tangerine (*Citrus deliciosa* Mediterranean) (CDM); orange (O); bitter orange (BO); tangerine (*Citrus reticulate* Owari) (CRO); pomegranate (P); non-pre-treatment (NP)).

The shortest drying time was 123 and 121 min for P pre-treated samples for 3 and 5 min dipping time, respectively (Figures 4 and 5). It reached to 164 min for CRO and 160 min for O pre-treated samples at same dipping times.

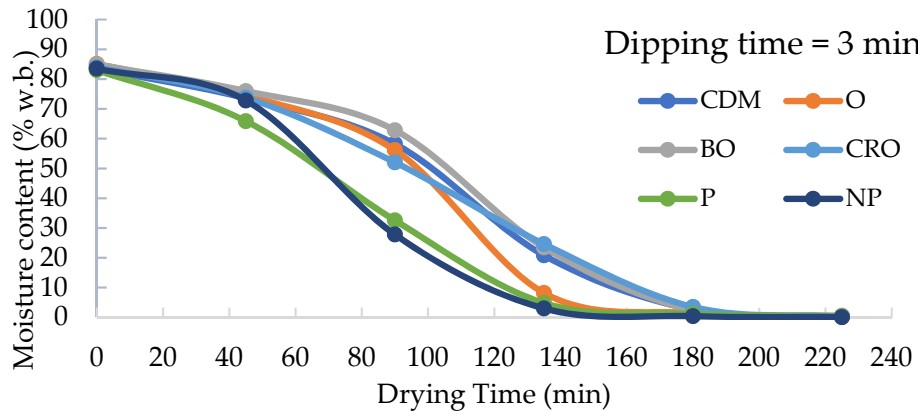

**Figure 4.** Effect of 3 min dipping time on drying time of different pre-treatments (tangerine (*Citrus deliciosa* Mediterranean) (CDM); orange (O); bitter orange (BO); tangerine (*Citrus reticulate* Owari) (CRO); pomegranate (P); non-pre-treatment (NP)).

The fastest dipping pre-treatment process for reaching the final moisture content of 10% (w.b.) was pomegranate juice application. Considering the drying times of the dipping times as 1, 3 and 5 min with pomegranate juice pre-treatment, it resulted in 135, 123 and 121 min, respectively.

Dehghannya et al. [48] found that increasing the osmotic solution dipping time enhanced water loss in convective and microwave drying for pre-treated quinces in a sucrose osmotic solution. This pre-treatment resulted in a quicker drying time. Drying time varied between 273 and 498 min and 278 and 585 min in ON and ON/OFF operating modes,

respectively. Doymaz et al. [49] studied the convective drying of quince slices pre-treated with citric acid solution or blanched in hot water. The results showed that pre-treatment with citric acid solution resulted in the fastest drying time. The blanched and untreated samples took 555 and 645 min to dry at 50 °C, respectively. This time was 540 min at the same temperature for citric acid solution pre-treated samples. Furthermore, blanched samples dried faster than untreated samples. Icier et al. [50] investigated the osmotic drying process after pre-treatments with electrical (60 V/cm, 15 s) and ultrasonic (195 W, 15 s) applications. Slices of samples were immersed in a solution of ascorbic acid (1%) and citric acid (0.2%) for 5 min. A saccharose solution [50% (*w/w*)] was used in the osmotic drying process. While electrically pre-treated samples took 240 min to dry, non and ultrasonic pre-treated samples took 300 min to reach a total dry matter rate of 40%. Following pre-treatments, Denge [33] studied multiple drying processes and discovered that there were no differences in drying periods amongst the pre-treatments. On the other hand, pre-treated samples were dried to lower final moisture content levels. There were statistically significant differences between pre-treatments in drying time. Thus, dipping material and dipping time affects the drying time, generally lowering the drying time.

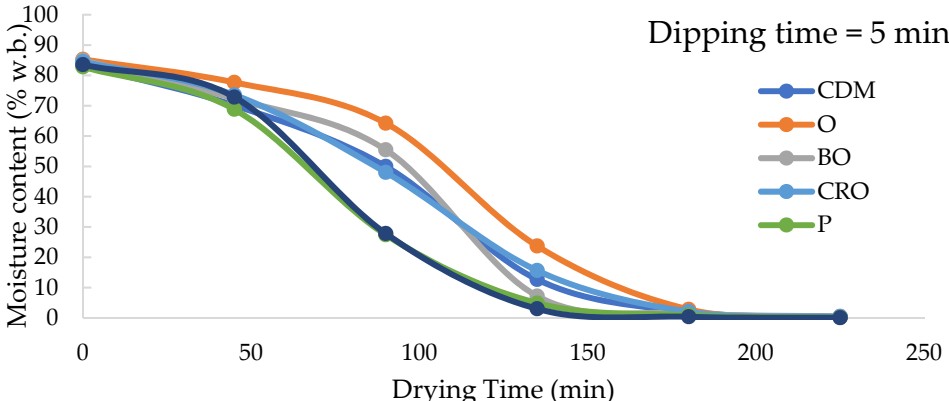

**Figure 5.** Effect of 5 min dipping time on drying time of different pre-treatments (tangerine (*Citrus deliciosa* Mediterranean) (CDM); orange (O); bitter orange (BO); tangerine (*Citrus reticulate* Owari) (CRO); pomegranate (P); non-pre-treatment (NP)).

### 3.4. Dipping Pre-Treatment Effect on Color and Biocompound Analysis

The color change of 3 mm thick quince slices at different drying air temperatures are given in Figures 6 and 7 and Table 3. As can be seen, while the C* and h° value did not change with the increase in the temperature, the TP and TEAC values increased linearly. However, the highest ΔE value (22.03) was determined at drying air temperature of 70 °C, and the lowest (12.04) at drying air temperature of 60 °C in non-pre-treatment. While the TF value was 120.74 μmol TE/g d.w. in fresh quince slice, it was found as 720 at 60 °C, 531 at 70 °C and 1445 μmol TE/g d.w. at 80 °C.

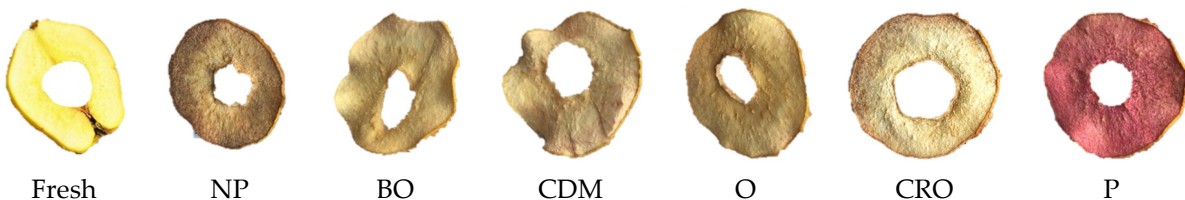

**Figure 6.** Fresh, non-pre-treated (NP) and dried in 1 min dipping time with different fruit juices. (tangerine (*Citrus deliciosa* Mediterranean) (CDM); orange (O); bitter orange (BO); tangerine (*Citrus reticulate* Owari) (CRO); pomegranate (P)).

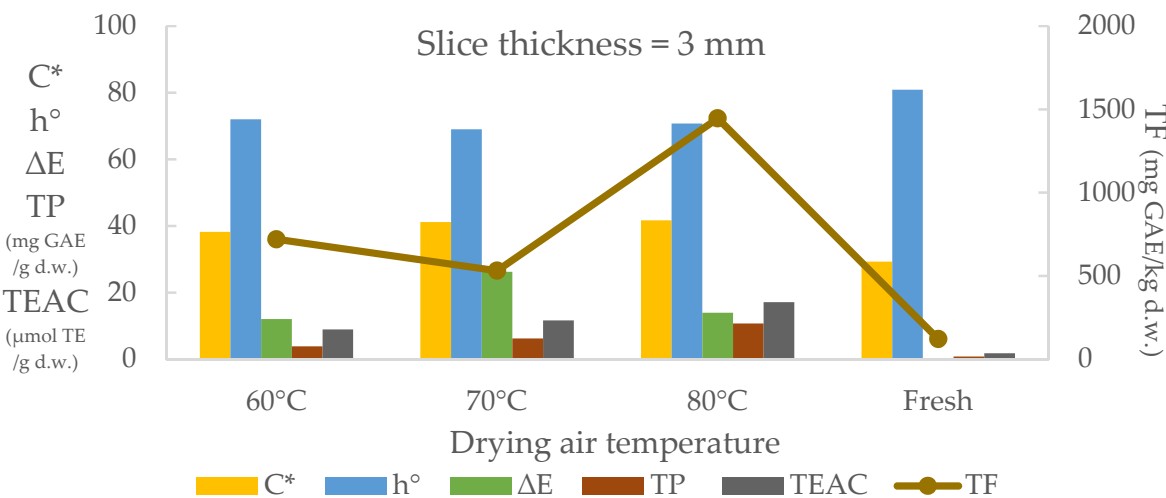

**Figure 7.** Effects of drying air temperature on quality parameters.

**Table 3.** Dipping Pre-treatment and Time Effect on Color and Bioactive Compounds Properties (tangerine (*Citrus deliciosa* Mediterranean) (CDM); orange (O); bitter orange (BO); tangerine (*Citrus reticulate Owari*) (CRO); pomegranate (P); non-pre-treatment (NP)).

| Dose | Dipping Time (min) | L* | a* | b* | C* | h° | ΔE | BI | TP (mg GAE /g d.w.) | TEAC (μmol TE /g d.w.) | TF (mg GAE /kg d.w.) |
|---|---|---|---|---|---|---|---|---|---|---|---|
| BO | 1 | 72.01 [b] | 8.33 [a] | 39.90 [a,b,*] | 40.77 [a] | 78.22 [b] | 12.16 [a] | 13.70 [a] | 10,800.44 [b] | 23.80 [b] | 1481.11 [b] |
|  | 3 | 73.27 [b] | 8.73 [a] | 39.25 [a] | 40.25 [a] | 77.40 [b] | 12.55 [a] | 13.79 [a] | 12,970.44 [c] | 26.21 [b] | 1644.07 [c] |
|  | 5 | 72.79 [b] | 8.58 [a] | 39.73 [a,b,*] | 40.71 [a] | 77.80 [b] | 12.71 [a] | 13.81 [a] | 14,589.19 [d] | 24.48 [b] | 2018.15 [d] |
| NP | - | 57.15 [a] | 15.77 [b] | 43.13 [b] | 45.98 [b] | 70.28 [a] | 22.03 [b] | 27.30 [b] | 6194.46 [a] | 11.63 [a] | 531.11 [a] |
| CDM | 1 | 68.61 [c] | 11.52 [a] | 43.18 [a] | 44.76 [a] | 75.32 [b] | 16.78 [a,b,*] | 18.42 [a] | 9006.69 [b] | 18.62 [b] | 1671.85 [b] |
|  | 3 | 62.35 [a,b,*] | 14.69 [b] | 50.74 [b] | 52.83 [b] | 73.85 [b] | 25.14 [c] | 24.69 [b] | 11,261.13 [c] | 24.21 [c] | 1757.04 [c] |
|  | 5 | 68.01 [b,c,*] | 10.22 [a] | 42.44 [a] | 43.66 [a] | 76.63 [b] | 16.20 [a] | 17.22 [a] | 16,777.80 [d] | 26.04 [d] | 1757.03 [c] |
| NP | - | 57.15 [a] | 15.77 [b] | 43.13 [a] | 45.98 [a] | 70.28 [a] | 22.03 [b,c,*] | 27.30 [b] | 6194.46 [a] | 11.63 [a] | 531.11 [a] |
| O | 1 | 67.14 [b] | 9.05 [a,b,*] | 40.25 [a,b,*] | 41.27 [a] | 77.40 [b] | 12.59 [a,b,*] | 15.47 [a] | 12,136.13 [c] | 21.38 [b] | 1807.03 [c] |
|  | 3 | 67.16 [b] | 7.34 [a] | 38.23 [a] | 38.94 [a] | 79.22 [b] | 10.19 [a] | 13.42 [a] | 12,077.80 [c] | 21.89 [b] | 2012.59 [d] |
|  | 5 | 68.67 [b] | 10.26 [b] | 44.59 [c] | 45.82 [b] | 77.02 [b] | 17.08 [b] | 17.10 [a] | 11,402.80 [b] | 21.26 [b] | 1482.96 [b] |
| NP | - | 57.15 [a] | 15.77 [c] | 43.13 [b,c,*] | 45.98 [b] | 70.28 [a] | 22.03 [c] | 27.30 [b] | 6194.46 [a] | 11.63 [a] | 531.11 [a] |
| CRO | 1 | 67.00 [b] | 15.13 [b] | 51.49 [b] | 53.68 [b] | 73.64 [b] | 25.12 [b] | 23.52 [a,b,*] | 8977.80 [b] | 18.50 [b] | 1155.18 [c] |
|  | 3 | 67.09 [b] | 11.59 [a] | 44.86 [a] | 46.35 [a] | 75.56 [b] | 17.79 [a] | 18.84 [a] | 8877.80 [b] | 18.70 [b] | 808.89 [b] |
|  | 5 | 70.66 [b] | 13.65 [a,b,*] | 50.26 [b] | 52.11 [b] | 75.01 [b] | 23.45 [b] | 20.71 [a] | 10,944.46 [c] | 19.18 [c] | 1221.85 [c] |
| NP | - | 57.15 [a] | 15.77 [b] | 43.13 [a] | 45.98 [a] | 70.28 [a] | 22.03 [a,b,*] | 27.30 [b] | 6194.46 [a] | 11.63 [a] | 531.11 [a] |
| P | 1 | 49.95 [b] | 32.55 [b] | 19.47 [a] | 38.12 [a] | 31.29 [b] | 35.23 [b] | 46.79 [b] | 12,438.91 [b] | 29.54 [b] | 1461.97 [c] |
|  | 3 | 45.24 [a] | 36.10 [c] | 16.11 [a] | 39.87 [a,b,*] | 23.88 [a] | 41.50 [c] | 53.66 [b] | 12,394.46 [b] | 32.17 [c] | 957.03 [b] |
|  | 5 | 41.19 [a] | 39.24 [c] | 18.89 [a] | 43.64 [b,c,*] | 25.62 [a] | 45.61 [c] | 63.15 [c] | 13,852.80 [c] | 32.49 [c] | 1886.66 [d] |
| NP | - | 57.15 [c] | 15.77 [a] | 43.13 [b] | 45.98 [c] | 70.28 [c] | 22.03 [a] | 27.30 [a] | 6194.46 [a] | 11.63 [a] | 531.11 [a] |

\* According to the Duncan multiple comparison results, the difference between the means with the same letter is insignificant.

In Figure 8 and Table 3, color analysis results of different quince slice thicknesses are given. There were no significant differences among C*, h° and TEAC values at different slice thicknesses. The biggest ΔE value was found for 3 mm thick quince slices. The highest TF, TP and TEAC value of 1440.37 μmol TE/g d.w., 12.54 mg GAE/g d.w. and 19.47 μmol TE/g d.w. were found in 6 mm thick quince slices, respectively.

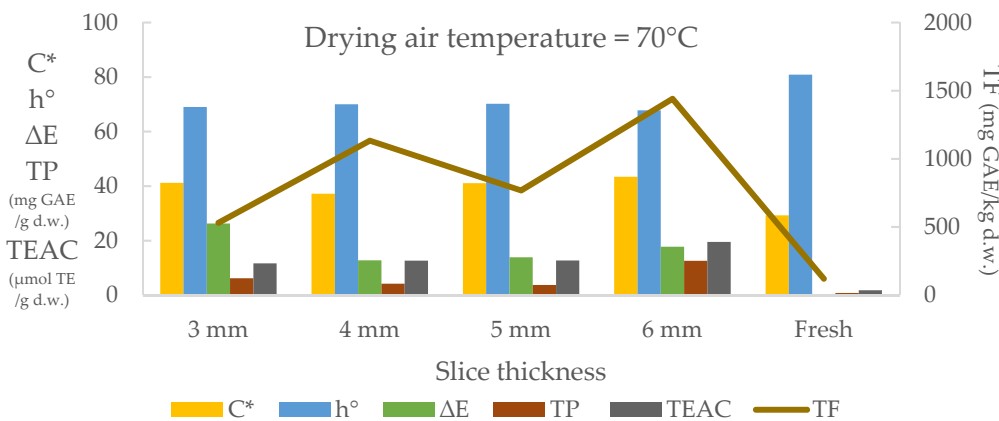

**Figure 8.** Effects of slice thickness on quality parameters.

As can be seen from Figure 9 and Table 3, BO, O and CDM pre-treatments had the lowest ΔE values in 1 min dipping times in different fruit juices, respectively, as 12.16, 12.59 and 16.78. While this situation remained the same at 5 min (Figure 10) of dipping time, it remained the same at 3 min (Figure 11) of dipping time except for CDM. The highest TEAC value was found in P pre-treatment, with values of 29.54, 32.17 and 32.49 µmol TE/g d.w., respectively, in all of the dipping processes in fruit juices at different dipping times.

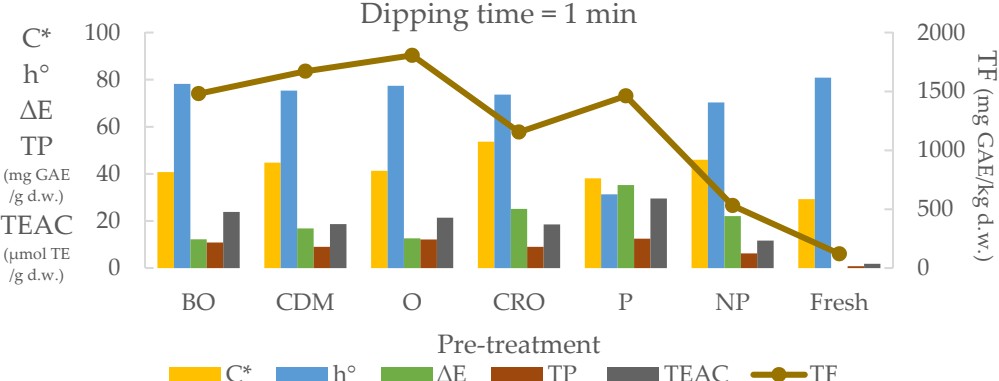

**Figure 9.** Effects of 1 min dipping time on quality parameters (tangerine (*Citrus deliciosa* Mediterranean) (CDM); orange (O); bitter orange (BO); tangerine (*Citrus reticulate* Owari) (CRO); pomegranate (P); non-pre-treatment (NP)].

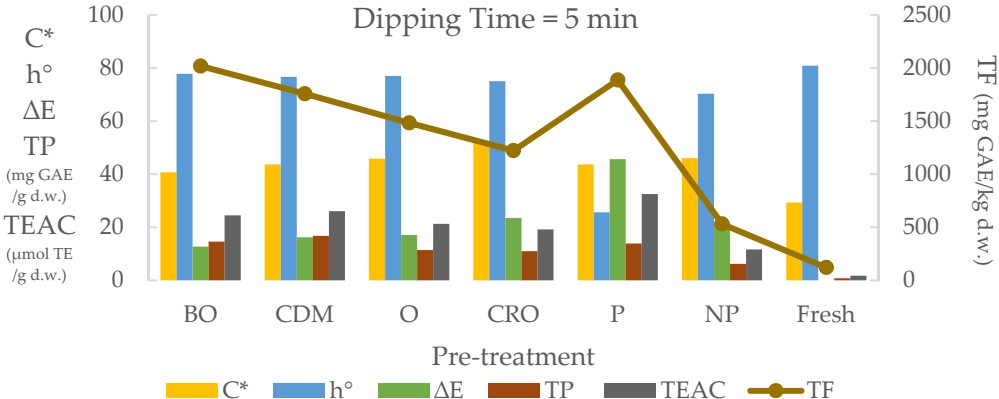

**Figure 10.** Effects of 5 min dipping time on quality parameters (tangerine (*Citrus deliciosa* Mediterranean) (CDM); orange (O); bitter orange (BO); tangerine (*Citrus reticulate* Owari) (CRO); pomegranate (P); non-pre-treatment (NP)).

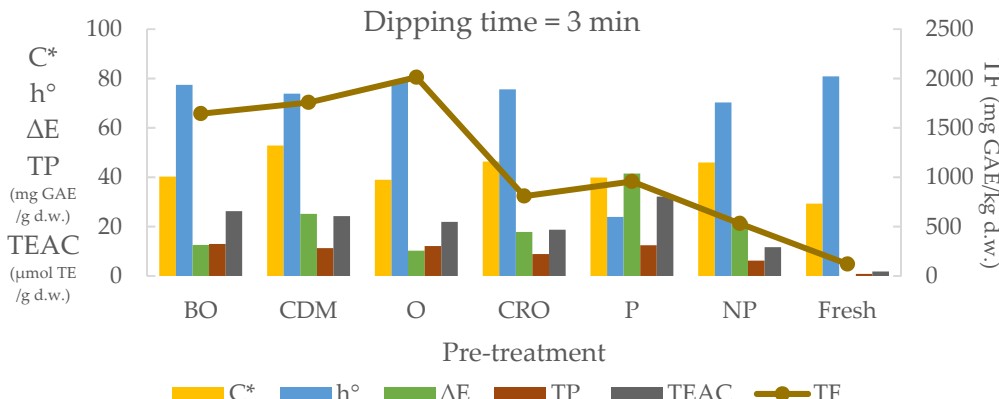

**Figure 11.** Effects of 3 min dipping time on quality parameters (tangerine (*Citrus deliciosa* Mediterranean) (CDM); orange (O); bitter orange (BO); tangerine (*Citrus reticulate* Owari) (CRO); pomegranate (P); non-pre-treatment (NP)).

Considering the results of the a* value, which is the redness value, the highest value was determined in the trial design, which was kept in pomegranate juice. The lowest value was found in the bitter orange trial design. It was determined that the brightness value L* and the redness value a* were inversely proportional to the test results. The trials with the highest ΔE value, which is the total color change, were observed in quince slices kept in pomegranate juice. According to the BI results, the values lower than the non-pre-treatment trials were in dipping juice and dipping time for 1 min in BO-CDM-O-P, for 3 min in BO-O-CRO-P and for 5 min in BO-CDM-O-P. As expected, the highest BI value was observed in quince slices kept in pomegranate juice. All juice dipping pre-treatments except P application increased the L* value. BO and O pre-treatment applications decreased the ΔE value compared to the non-pre-treated samples.

Kalejahi and Asefi [51] found that raising the infrared radiation power level enhanced the L* and b* values while decreasing the a* value in infrared drying. The best results were obtained at an 800 W power level with the lowest ΔE value. Variable drying methods resulted in significant color changes, according to Elmizadeh et al. [52]. According to the findings, the L* value decreased after drying in all pre-treatments. Increasing the temperature in convective drying had no effect on the L* value. According to the statistical data, however, increasing temperature had no effect on a* and b* values in convective drying. At 70 °C, dried quince slices showed lower BI values than at 50 °C and 60 °C, but at 50 °C and 60 °C, they were near. At 50, 60 and 70 °C, the quince slices had BI values of 262.55, 260.40 and 252.02, respectively. As a result, the drying procedure had a considerable impact on the BI value with mean BI values of 252.02, for EHD and convective drying dried quince slices. The color quality of the hot-air dried quince slices was superior to that of EHD drying. When dried by osmotic technique to a total solid weight of 40%, Icier et al. [50] discovered that both ultrasonic and non-pre-treated samples differed in terms of L*, a*, hue angle, and ΔE. The L* values of non-ultrasonic and ultrasonic pre-treated samples were similar, whereas the L* values of electrically pre-treated samples were substantially different. The L* value of ultrasonic pre-treated samples climbed in non-pre-treated samples after osmotic drying, but declined in electrically pre-treated samples. When compared to non-treated and ultrasonic-treated samples, electrical pre-treatment resulted in a higher level of redness. After pre-treatment, the b* value of osmotically dried samples did not change considerably. Pre-treatments before osmotic drying had no effect on the chroma value of quince slices. While there were no changes in ΔE between ultrasonic and non-pre-treated samples, electrically pre-treated samples had differences. Pre-treatment affected the color of quince slices in general. Before osmotic drying, electrically pre-treated quince slices had different L*, a*, b*, and hue angles than non and ultrasonic pre-treated samples using a microwave convective dryer. Taghinezhad et al. [53] dried quince slices (4 mm) at various power levels, temperatures and velocities. According to the findings, ΔE for quince slices

ranged from 10.85 to 35.18 under various drying conditions. Lower drying air temperature, microwave power and air velocity levels, in other words, resulted in fewer color variations. Turkiewicz et al. [44] studied the effects of several drying processes on the color of quince slices (3 mm) as color, including convective, vacuum–microwave, a combination and freeze drying. Darkening was discovered in all fresh fruit drying techniques, according to the findings. The vacuum–microwave drying at 480 W and freeze drying had the lowest L* values. Vacuum–microwave drying (480/120 W and 120 W) produced similar outcomes and natural-colored items. In all drying techniques, the value of a* rose towards the red color, similar to the L* value. Vacuum–microwave drying at 480 W yielded the largest rise of almost six times. Freeze drying produced the smallest changes in a* value, as it did in L* value. The proportion of green color was reduced by convective and vacuum-microwave drying. The b* value increased in all dried quince slices obtained. The b* value decreased as the temperature in convective drying and the power level in vacuum-microwave drying were increased. The lowest color changes were generated by freeze drying. Izli and Polat [13] discovered that the drying methods had an effect on color, with freeze drying being the closest to fresh. The L* values of the dried samples declined dramatically from 79.7 to 60.8 at 75 °C, according to the results. The freeze drying procedure had the greatest L* value (71.87), whereas the L* values of the other samples were closer to those of the fresh samples. The hues of the samples dried at 75 °C were darker than the freeze dried ones. It was discovered that when the temperature rose, the L* values dropped. In all drying procedures, the fresh sample's redness value (a*) increased dramatically. For convective drying, the a* values of the samples ranged from 7.91 to 10.39. The freeze dried samples' a* values were the most similar to those of fresh samples. At 75 °C, all samples' b* values dropped dramatically from 37.75 to 31.31. The freeze dried samples had the greatest b* value of 37.75. The lowest b* value was found in samples treated at 75 °C (31.31). The color angle values of the dried quince cubes were much higher than those of the fresh samples. All of the dried samples' hue values were lower than the fresh sample's. This indicated that browning was more prevalent in the dried samples. In all dried samples, the chroma values of the fresh samples fell, similar to hue values. In contrast to convective drying, freeze drying produced the best product color outcomes, which were the most similar to the L* and b* values of fresh samples. Celen and Kus [37] found that, the microwave power level of 2000 W and belt velocity of 0.210 m/min produced the greatest drop in L* value. At the same belt speeds, studying at 2800 W produced colors that were closer to the fresh ones. Color values were improved as microwave power was increased. The best colors were obtained with a belt velocity of 0.175 m/min and a power level of 2000 W, according to the data. Increased oven temperature and vacuum oven drying procedure increased color changes, causing browning, according to Denge [33]. Furthermore, pre-treatment with sugar solution resulted in caramelization, lowering the L* value. The total color changes were likewise reduced as the vacuum pressure was increased. The total color change was the lowest in sulfite-treated quince samples, whereas it was highest in sugar-treated quince samples. The ascorbic acid-treated quince slices show the most total color change. The total color change at various microwave power levels was obtained, indicating that raising the microwave power level affected the overall color. Burnt samples and significant color changes were observed, particularly at microwave power levels of 750 W. In both pre-treated and non-pre-treated samples, microwave dried quince samples changed color more than the other procedures. The complete color change was similar in the oven and vacuum oven at the same drying temperatures. The color shift was the smallest when the vacuum pressure was increased at the same temperature. Except for the pre-treatment with sugar solutions, there was virtually little difference in complete color change when microwave drying at 100 and 180 W power levels was compared. Chemically pre-treated quince samples showed the least color change, while ethyl oleate + sodium meta bisulfite solution produced the best colored samples.

When the CDM pre-treatment application was examined, the highest TEAC was observed in 5 min of dipping time, while the lowest was observed in 1 min after the

application without pre-treatment (Table 3). The highest TF value was found in BO for 5 min (2018.15 mg GAE/kg d.w.) and O for 3 min (2012.59 mg GAE/kg d.w.) pre-treatments. Except for the O and CRO pre-treatments, the TP value increased as the dipping time increased in the other trials. The highest TP value was reached after 5 min dipping of CDM.

For diverse quince varieties and genotypes, Wojdylo et al. [54] studied the phenolic components, L-ascorbic acid, antioxidant and PPO activity. The total polyphenol content ranged from 1709.43 mg/100 g dry weight (genotype 'S1' variety) to 3436.56 mg/100 g dry weight Leskova variety. The effects of quince variety on the antioxidant activity measured by ABTS ranged from 2.4 to 0.9, 2.5 to 0.9 for FRAP, and 1.5 to 0.4 for TEAC technique. Szychowski et al. [36] found total phenolic compounds followed the same rise as phenolic acids, with the mean of all samples reaching 5183 mg $(100 \text{ g dm})^{-1}$, according to the findings. From the greatest to the lowest antioxidant activity values, the treatments were as follows: FD > VMD > CDCPD-VMFD. Within the VMD, those with the best antioxidant activity behavior were VMD at 480 W, followed by VMD at 120 W. CD and CPD-VMFD dry samples had the lowest antioxidant capacity. In microwave and convective drying, Baltacioglu et al. [15] investigated the effects of slice thickness, microwave power and drying air temperature on antioxidant activity and total phenolic content of quince. Maximum antioxidant activity was found at 285 W and 4.14 mm thickness in microwave drying, while maximum total phenolic content was obtained at 285 W and 6.85 mm thickness. The highest antioxidant activity and total phenolic content were achieved in convective drying at 75 °C and 1.2 mm thickness. After convective drying, the antioxidant activity of quince slices decreased. The antioxidant activity was influenced by sample thickness and temperature. At varied drying temperatures, increasing the thickness from 2 to 4 mm resulted in reduced antioxidant activity. Increase in thickness and power level in microwave drying resulted in a drop in total phenolic content, followed by a rise in antioxidant activity. Total phenolic content and antioxidant activity were determined to be 1544 mg GAE/1000 g and 74 percent, respectively, after natural sun drying. These values were similar to those obtained after microwave drying and were greater than those obtained after convective drying. Izli and Yildiz [55] examined the impact of high-intensity ultrasound (HUS) pre-treatment on some quality characteristics of convective-dried (CVD) quince fruit pieces at different temperatures (50, 60, 70 and 80 °C). The lowest antioxidant capacity (AOC) was determined for dried quince samples at 50 °C with no HUS pre-treatment (2.58 μmol TE/g d.w.), the highest AOC was determined for the dried quince samples exposed to HUS pre-treatment for 15 min at 70 °C (7.08 μmol TE/g d.w.). Turkiewicz et al. [41] investigated the effects of different drying methods on the quality characteristics of quince slices (3 mm), such as phenolic chemicals and antioxidant activity: convective, vacuum–microwave, combination (convective + vacuum + microwave) and freeze drying. Vacuum microwave drying yielded the highest antioxidant activity levels. The antioxidant activity levels declined with decreasing microwave power in vacuum microwave drying and convective drying with long exposure to hot air, with the lowest values recorded for combination drying. At the same time, vacuum microwave drying of quince slices produced the greatest value, whereas convective drying and convective vacuum microwave drying at 50 °C produced the lowest. In addition, depending on the drying process used, the total phenol concentration decreased by 10–33%. The most significant alterations occurred during convective drying, while the lowest happened after freeze drying. Dried fruits dried using convection at 70 °C showed a higher total phenolic content than those dried at 50 °C, owing to the 2 h longer drying time at a lower temperature. When compared to convective and vacuum microwave drying individually, convective vacuum microwave drying better preserved the color and important bioactive components. Sohrabpour et al. [56] found the highest total phenolic content as 57.29 mg GAE/mL for ascorbic acid pre-treated quince slices.

## 4. Conclusions

The fastest drying was achieved in pomegranate pre-treated quince slices for 5 min dipping time in 121 min. The slowest drying was obtained in 168 min with BO application for 1 min dipping time.

The difference in total color, ΔE, was the highest in pomegranate juice dipping pre-treatment. BI value was found high for pomegranate pre-treatment due to its natural dark color. The red value a* was found to be higher than the non-pre-treated sample in the 5–3 and 1 min dipping time of the pomegranate, respectively, as expected. Quince slices with the highest C* value were found for tangerine (*Citrus reticulata* Owari) and tangerine (*Citrus deliciosa* Mediterranean) juice dipping pre-treatments. The group with the lowest L* value was found in quince slices that were pre-treated in pomegranate juice for 5 min dipping time. In quince slices dipped in pomegranate juice for any dipping time, BI and a* values were the highest, while L*, b* and h values were the lowest.

The lowest TP, TEAC and TF values were observed in fresh and non-pre-treated dried quince slices. This showed that all the applied pre-treatments had a positive effect on the biocompounds of quince slices. Among the TP and TEAC values, the most effective pre-treatment was determined as P. Considering the TF values, it was determined that BO, CDM and O applications were the most effective pre-treatment.

It was observed that the drying time, slice thickness, drying temperature and pre-treatments directly affect the drying characteristics, and the determination of the optimum drying parameters was important for the product quality and dipping time of the dried product. Therefore, pomegranate dipping at a drying air temperature of 70 °C and slice thickness of 3 mm resulted in shorter drying time with higher TP and TEAC values.

**Author Contributions:** Conceptualization, H.E.A., I.B. and C.E.; methodology, H.E.A., I.B., O.S., S.G. and C.E.; software, H.E.A. and I.B.; investigation, C.E. and S.G.; resources, C.E. and O.S.; writing—original draft preparation, H.E.A., I.B., C.E., O.S. and S.G.; writing—review and editing, C.E., H.E.A. and I.B.; visualization, C.E.; supervision, C.E. All authors have read and agreed to the published version of the manuscript.

**Funding:** This research received no external funding.

**Institutional Review Board Statement:** Not applicable.

**Informed Consent Statement:** Not applicable.

**Acknowledgments:** The authors are grateful to Kamil Ekinci, Isparta Applied Science University, Faculty of Agriculture, for English language editing.

**Conflicts of Interest:** The authors declare no conflict of interest.

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
