# Peer review of "Effects of Convective Drying of Quince Fruit (Cydonia oblonga) on Color, Antioxidant Activity and Phenolic Compounds under Various Fruit Juice Dipping Pre-Treatments"

_agriculture, doi:10.3390/agriculture12081224_

Round 1

Reviewer 1 Report

This study investigated some characteristics of dried quince slices dipped in several types of fruit juice. The structure and the research results are well presented and then discussed. But, statistical analysis is not observed in it.

In my opinion, the present manuscript is suitable for publication in Agriculture journal after the following corrections:

L20: Use other keywords that are not in the title of the article.

L64-66: Please add the ultrasound and radiation as methods of drying and add the following two references in the list of references:  

Roueita, G., Hojjati, M., & Noshad, M. (2020). Study of physicochemical properties of dried kiwifruits using the natural hypertonic solution in ultrasound-assisted osmotic dehydration as pretreatment. International Journal of Fruit Science20(sup2), S491-S507.

Salehi, F., Kashaninejad, M., & Jafarianlari, A. (2017). Drying kinetics and characteristics of combined infrared-vacuum drying of button mushroom slices. Heat and Mass Transfer53(5), 1751-1759.

 L71: Write the different dipping pretreatments (fruit juices).

L74: write Cydonia oblonga in italic form.

L74-75: What season of the year were the fruits prepared?

L80: "… different fresh fruit juices.." is correct.

L85 or L 92: Write the technical specifications and manufacturer of the cabinet dryer in full.

L105: Write the complete specifications of the device manufacturer.

- Statistical analysis method should be written at the end of the Materials and Methods section. The titles of all figures, especially figure 6 and 7, are not complete and need to be written more completely. Also, How many times have the tests been repeated?

- In Figure 2, the drying time of the sample is continued until the moisture level reaches zero percent, while in the text of the article (L147) it is mentioned that the samples are dried until the moisture level reaches ten percent. Explain more about it.

- In Table 2, the difference between the treatments should be shown statistically by placing English lowercase letters in each column.

- The captions of all figures, especially figures 6 and 7, are not complete and need to be written more completely.

- Put the Error Bars on the columns of the figures.

-Does the use of fruit juice as a pre-treatment affect the sensory properties of dried quinces? Were the sensory characteristics of the treatments such as taste, odor, and texture investigated?

L469: At the end of the paragraph, the general conclusion and suggestion or suggestions from this research should be written.

Author Response

Please find it in attached pdf file.

Reviewer 2 Report

1.      The authors only wrote 3mm thickness quince in the abstract, but in the method, there are 4 thickness levels

2.      In sample preparation, please add the physical properties of the quince used such as harvest age, dimensions, and color

3.      Please provide the details of all measurement instruments used, e.g. electric balance, the thermometer, the hygrometer, food mixer, etc.

4.      Please provide detailed information about the preparation of bitter orange juice, 2 different tangerine juice, orange juice, pomegranate juice, the volume of dipping solution, the physical properties of bitter orange, 2 various of tangerine, orange and pomegranate and also the volume of each juice before dipping.

5.      Please provide detailed information and equation about the statistical analysis on method chapter.

6.      At “Drying test” section, there are no reference explaine about the standard method of measuring water content (temperature and time).

7.      Why the drying time of 3mm thickness quince at Figure 1 and 2 are very different?

8.      At Figure 2, why author only explain the drying curve at 70°C drying temperature? What about at 60 and 80°?

9.      The authors must explain explicitly and give the reason how the effect of dipping time on drying time, on color and biocompound. Not only citing from other journal.

10.   The image of dried samples from each treatment must be provided

11.   The authors mention about Lref, aref and bref. Please provide detailed value of those parameters. At Table 2, the authors only show the L*, a*, b* and C*

12.   What is the best treatment and drying temperature based on this reasearch?

Author Response

Please find it in attached pdf file.

Reviewer 3 Report

This article works on convective drying and determining aspects such as drying temperature, thickness and different juices dipping as pre-treatment. 

Comments as below:

1. Introduction: To add the problem statement and rationale of convective drying, dipping of fruit juices. Also, add on the background on color and antioxidant etc influenced by drying temperature, thickness, and juice dipping

2. Sample preparation- scientific name should be italic. On what basis that the slices are 3-6 mm? any references?

3. Pre-treatment of samples- to add in the pH of the juices (table 1)

4. To add references for Cabinet convective dryer and drying test (methods)

5. Effect of drying air temperature- rephrase/ clarify the sentence (reaching to safe moisture) and drying time was found statistically important (same as last sentence of page 5)

6. In the discussion section, there are alot of discussion point quoted from others using microwave, vacuum etc. Remove as it is not supporting the investigation of this study which focus on convective drying. Support with suitable references. 

7. Figures- to add error bar and use different style for line. 

8. For discussion, include support that explains why thickness increase the drying time; and why different dipping influence the moisture etc. 

9. Table 2. to have footnote that describes the abbreviation. Also, choose either table 2 OR figure 6-9.

10. For page 11. separate into shorter paragraph and rearrange the disucssion based on L, a, b and total color changes. Dont mix the results and discussions points in 1 big paragraph. Each paragraph or two should focus on L and then move on to a, b etc, respectively. 

3. 

Author Response

Please find it in attached pdf file.

Round 2

Reviewer 1 Report

It is necessary to write the method of statistical analysis (Statistical Analyses) and the specifications of the software used (for example, SPSS or SAS, etc.) along with the method of comparing averages (for example Duncan, ...)  at the end of the materials and methods section and before the results and discussion section.

Please read the papers published in this journal for more guidance.

Author Response

Find our comments attached.

Reviewer 2 Report

1.      In sample preparation, please add the physical properties of the quince, bitter orange, 2 different tangerine, orange and pomegranate such as dimensions (diameter, length) in average and color.

2.      Please provide detailed information about the volume of each juice for dipping treatment.

3.      Please provide detailed information and equation about the statistical analysis on method chapter. Please mention table number.

4.      At Figure 2, why author only explain the drying curve at 70°C drying temperature? What about at 60 and 80°? Author should explain the drying curve at 60 and 80°

5.      The image of dried samples from each treatment must be provided. At least fresh and final dried quinces

Author Response

Find our comments attached.

Reviewer 3 Report

Majority of the comments addressed accordingly. 

Author Response

Find our comments attached.
